# Central Lymph Node Ratio Predicts Recurrence in Patients with N1b Papillary Thyroid Carcinoma

**DOI:** 10.3390/cancers14153677

**Published:** 2022-07-28

**Authors:** Il Ku Kang, Kwangsoon Kim, Joonseon Park, Ja Seong Bae, Jeong Soo Kim

**Affiliations:** Department of Surgery, College of Medicine, The Catholic University of Korea, Seoul 06591, Korea; kik5304@gmail.com (I.K.K.); noar99@naver.com (K.K.); joonsunny@naver.com (J.P.); btskim@catholic.ac.kr (J.S.K.)

**Keywords:** lymph node ratio, papillary thyroid carcinoma, N1b, disease-free survival

## Abstract

**Simple Summary:**

The lymph node ratio (LNR) is an emerging predictive marker for recurrence in papillary thyroid carcinoma (PTC). The purpose of this study was to investigate the association between LNR and disease-free survival (DFS) in patients with N1b PTC. Unlike that in the lateral or whole neck, LNR in the central compartment (CLNR) was found to have prognostic significance. The high-CLNR group (CLNR ≥ 0.7) had worse DFS and was 4.5 times more likely to experience recurrence in patients with N1b PTC.

**Abstract:**

The lymph node ratio (LNR) indicates the number of metastatic lymph nodes (LNs) to the total number of LNs. The prognostic value of LNR in papillary thyroid carcinoma (PTC) and other solid tumors is known. This study aimed to investigate the relationship between LNR and disease-free survival (DFS) in patients with PTC with lateral LN metastases (N1b PTC). A total of 307 patients with N1b PTC who underwent total thyroidectomy and therapeutic central and lateral LN dissection were retrospectively analyzed. The DFS and recurrence risk in the patients with LNR, central-compartment LNR (CLNR), and lateral-compartment LNR (LLNR) were compared. The mean follow-up duration was 93.6 ± 19.9 months. Eleven (3.6%) patients experienced recurrence. Neither LNR nor LLNR affected the recurrence rate in our analysis (*p =* 0.058, *p =* 0.106, respectively). However, there was a significant difference in the recurrence rates between the patients with low and high CLNR (2.1% vs. 8.8%, *p =* 0.017). In the multivariate analysis, CLNR ≥ 0.7 and perineural invasion were independent predictors of tumor recurrence. High CLNR was associated with an increased risk of recurrence, and was shown to be a significant predictor of prognosis in patients with N1b PTC.

## 1. Introduction

Papillary thyroid carcinoma (PTC) is the most common pathological type and accounts for approximately 90% of thyroid cancer [1]. Among all types of malignancies, PTC has a relatively good prognosis, with an estimated 10-year disease-specific survival of >96% [2,3]. In most cases, surgical treatment is sufficient. Depending on the risk of recurrence, such as that associated with size [4], extrathyroidal extension (ETE) [5], vascular invasion [6], or lymph node (LN) metastasis [7], either thyroid lobectomy or total thyroidectomy is performed. Approximately 30–80% of patients with PTC develop cervical LN metastasis [8]. PTC progresses from the thyroid gland to the adjacent central compartment and then to the ipsilateral and contralateral lateral neck compartments [9].

In the eighth version of the American Joint Committee on Cancer/Union for International Cancer Control (AJCC/UICC) tumor–node–metastasis (TNM) staging system, patients with PTC with lateral neck LN metastasis are classified as N1b. Whether PTC invades the LNs outside the carotid artery differentiates N1b from N1a, regardless of the number or size. The definition of N1b included superior mediastinal node metastasis (level VII) in the past, which was reclassified from N1b to N1a in the current edition. The surgical extent of patients diagnosed with N1b PTC includes therapeutic central and lateral LN dissection, as well as bilateral thyroid glands [10,11,12,13]. In the seventh version of the AJCC/UICC TNM staging system, patients with PTC with LN metastasis of 45 years or older were classified as stage III or Iva, depending on the location of metastasis (N1a or N1b). In the eighth version, however, they were classified into the same stage, as long as the tumor is limited to LNs in the neck [14]. Thus, the N category in the AJCC/UICC TNM staging is insufficient to evaluate the risk of recurrence in patients with node-positive PTC.

The American Thyroid Association (ATA) proposed a risk-stratification system with three categories: low, intermediate, and high, to predict the risk of recurrence [15]. Although the system recommends that the number and size of metastatic LNs be considered as risk factors to predict recurrence, the system lacks evidence to provide appropriate information for following up with patients with N1b PTC after surgical treatment.

The lymph node ratio (LNR), which is defined as the number of positive LNs divided by the total number of LNs harvested, is used to evaluate oncological prognosis in solid tumors, such as those in lung, gastric, and colon cancer [16,17,18]. In terms of PTC, David et al. reported, after analyzing 10,955 cases, that LNR ≥ 0.42 was associated with disease-specific mortality [19]. Recently, it has been suggested that LNR is a predictor of recurrence in patients with N1b PTC. Lee et al. suggested that LNR > 0.25 in the lateral compartment is an independent prognostic factor affecting recurrence [20]. Another study demonstrated that lateral LNR > 0.3 had a significant effect on cancer-specific mortality [21]. To the best of our knowledge, only a few studies on patients with N1b PTC have investigated the relationship between LNR and tumor recurrence.

The study’s aim was to investigate the relationship between LNR and DFS, determine an optimal cutoff for LNR, and validate the clinical significance as a predictor of tumor recurrence in patients with N1b PTC after surgical treatment.

## 2. Materials and Methods

### 2.1. Patients

We retrospectively reviewed 312 patients who had been diagnosed with N1b PTC and had undergone a total thyroidectomy with central and lateral neck dissection between January 2012 and December 2017 at Seoul St. Mary’s Hospital (Seoul, Korea). After excluding two patients who had not undergone lateral neck dissection as an initial surgery and three with distant metastasis at initial presentation, a total of 307 patients were enrolled in this study. The mean follow-up duration was 93.6 ± 19.9 months (range: 52–123 months). This retrospective cohort study was conducted in accordance with the Declaration of Helsinki (as revised in 2013) and approved by the Institutional Review Board of Seoul St. Mary’s Hospital, The Catholic University of Korea (IRB No: KC22RISI0318). The requirement for informed consent was waived due to the retrospective nature of this study.

### 2.2. Postoperative Management and Follow-Up

All enrolled patients were diagnosed and treated according to the ATA management guidelines [15]. The patients had undergone a physical examination, neck ultrasound (US), and serum thyroid function testing at 3–6-month intervals and annually thereafter. All patients had been regularly followed up by physical examination, thyroid function testing, serum thyroglobulin and anti-thyroglobulin antibody concentration measurements, and neck US at 3–6-month intervals and annually thereafter. Radioactive iodine (RAI) ablation was performed at 6–8 weeks postoperatively, and whole-body scans were performed 5–7 days after RAI ablation. Patients with suspected recurrence after neck US were confirmed by cytological examination using US-guided fine-needle aspiration during the routine follow-up evaluations.

### 2.3. Statistical Analysis

Continuous variables are presented as the mean ± standard deviation and categorical variables are presented as the number with percentage. Student’s t-test and Pearson’s chi-square test or Fisher’s exact test were used to compare continuous and categorical variables, respectively. Receiver operating characteristic (ROC) curve analysis was performed to determine the cutoff value for LNR relevant to the disease-free survival (DFS). We calculated the area under the curve (AUC), sensitivity, specificity, and 95% confidence interval (CI). The same procedure was repeated for LNR in the central (CLNR) and lateral (LLNR) compartments. Univariate and multivariate Cox regression analyses were performed to validate the predictors of DFS, with the hazard ratio (HR) and CI presented. A *p*-value of <0.05 was accepted as indicative of statistical significance. IBM SPSS Statistics for Windows, version 24.0 (IBM Corp., Armonk, NY, USA), was used to perform all statistical analyses.

## 3. Results

### 3.1. Optimal Cutoff Values Determined by ROC Curve Analysis

The results of ROC curve analysis for LNR, CLNR, and LLNR are shown in Figure 1. The cutoff values that we adopted as best predictors for tumor recurrence in patients with N1b PTC were LNR of 0.32 (AUC, 0.631; sensitivity, 0.545; specificity, 0.828; 95% CI, 0.438–0.823; and *p =* 0.141), CLNR of 0.7 (AUC, 0.680; sensitivity, 0.545; specificity, 0.791; 95% CI, 0.497–0.863; and *p =* 0.043), and LLNR of 0.16 (AUC, 0.620; sensitivity, 0.636; specificity, 0.625; 95% CI, 0.455–0.785; and *p =* 0.177), respectively.

### 3.2. Comparison of Baseline Clinicopathological Characteristics According to LNR

The results for LNR with a cutoff of 0.23 are presented in Table 1. The patients in the high-LNR group were significantly younger (40.6 ± 14.1 vs. 44.8 ± 12.6 years, and *p =* 0.006) and had larger tumors (1.8 ± 1.1 cm vs. 1.5 ± 1.0 cm, and *p =* 0.004) than those in the low-LNR group. Higher rates of lymphatic (69.7% vs. 85.3%, *p =* 0.002) and vascular (4.5% vs. 12.4%, *p =* 0.016) invasion were observed in the high-LNR group than in the low-LNR group. There were no significant differences in sex, multifocality, bilaterality, gross extrathyroid extension (ETE), perineural invasion, T category, and TNM stages. The number of positive LNs was significantly higher (17.9 ± 9.3 vs. 8.0 ± 4.7, *p <* 0.001) in patients with high LNR, but the number of harvested LNs was higher in the low-LNR group (59.3 ± 22.6 vs. 53.7 ± 22.9, *p =* 0.033). Three (1.7%) patients in the low-LNR group and eight (6.2%) in the high-LNR group had tumor recurrence. There was no significant difference in recurrence between the two groups (*p =* 0.058).

### 3.3. Comparison of Baseline Clinicopathological Characteristics According to CLNR

Table 2 shows a comparison of the baseline clinicopathological characteristics according to the CLNR. The rate of recurrence was higher in the high-CLNR group than that in the low-CLNR group (8.8% vs. 2.1%, *p =* 0.017). The cutoff value was 0.7.

There were fewer female patients in the high-CLNR group than in the low-CLNR group (48.5% vs. 71.1%, *p =* 0.001). The tumor size was larger in the high-CLNR group than in the low-CLNR group (2.0 ± 1.2 cm vs. 1.5 ± 1.0 cm, *p =* 0.001). There were no significant differences in the multifocality, bilaterality, gross ETE, BRAF positivity, lymphovascular invasion, perineural invasion, number of harvested LNs, T category, and TNM stages. The number of positive LNs was higher in the high-CLNR group than in the low-CLNR group, regardless of the neck compartment (18.6 ± 10.7 vs. 10.3 ± 6.7, *p <* 0.001; 10.6 ± 6.9 vs. 4.6 ± 4.1, *p <* 0.001; and 8.1 ± 5.8 vs. 5.7 ± 4.3, *p <* 0.001, respectively).

### 3.4. Comparison of Baseline Clinicopathological Characteristics According to LLNR

The baseline clinicopathological characteristics depending on the LNR in the lateral compartment are described in Table 3. A LLNR of 0.16 was adopted as the cutoff value to compare the factors related to tumor and node characteristics. There were no significant differences in age, sex, tumor size, gross ETE, perineural invasion, BRAF mutation, number of harvested LNs from the central neck, T category, and TNM stages. Patients in the high-LLNR group had more multifocal (67.9% vs. 52.3%, *p =* 0.008) and bilateral tumors (45.5% vs. 28.7%, *p =* 0.004) than the patients in the low-LLNR group. Between the low- and high-CNLR groups, the rates of lymphatic invasion (70.8% vs. 85.7%, *p =* 0.003) and vascular invasion (5.1% vs. 12.5%, *p =* 0.027) significantly differed. There was no significant difference in recurrence according to LLNR (2.1% vs. 6.3%, *p =* 0.106).

### 3.5. Univariate and Multivariate Analyses of the Risk Factors for Recurrence

Table 4 presents the analysis results of univariate and multivariate Cox regression to identify risk factors related to tumor recurrence. The tumor size (HR, 1.707; *p =* 0.005), vascular invasion (HR, 4.320; *p =* 0.031), perineural invasion (HR, 5.588; *p =* 0.011), number of positive LNs in the whole neck (HR, 1.048; *p =* 0.017), number of positive LNs in the lateral compartment (HR, 1.095; *p =* 0.037), high CLNR (HR, 11.026, *p =* 0.031), and CLNR ≥ 0.7 (HR, 4.238; *p =* 0.017) were found to be significant predictors of recurrence in univariate analysis. In multivariate analysis, perineural invasion (HR, 6.045; *p =* 0.008) and higher CLNR with a cutoff of 0.7 (HR, 4.451; *p =* 0.014) were independent factors for predicting tumor recurrence. In Kaplan–Meier analysis, there was a statistically significant difference in DFS between the high- and low-CLNR groups (log-rank *p =* 0.009; Figure 2).

### 3.6. Recurrence Patterns for the Study Population

The recurrence patterns of the study cohort are summarized in Table 5. Three patients in the low-CLNR group experienced recurrence in the ipsilateral lateral compartment. Only one patient developed recurrence in the contralateral lateral compartment (level III LNs). In contrast, three out of six patients in the high-CLNR group developed recurrence in the contralateral lateral compartment. Patient No. 1 experienced recurrence in the left-level-5 LNs after bilateral lateral neck dissection diagnosed with bilateral metastasis. The largest tumor (5.0 cm) found during primary surgery was located in the left lobe of the thyroid gland. The recurrence of Patient No. 5 was found in the left-level-6 (central) LNs, which was ipsilateral to the largest tumor (2.7 cm) in the thyroid gland.

## 4. Discussion

Our results showed no significant difference in the recurrence rates between the low-LNR and high-LNR groups divided by an LNR cutoff value of 0.23 (1.7% vs. 6.2%, *p =* 0.058) among a total of 307 patients with N1b PTC. With respect to CLNR, however, the rate of tumor recurrence differed significantly between the high- and low-CLNR groups (8.8% vs. 2.1%, *p =* 0.017). CLNR ≥ 0.7 was an independent prognostic factor for recurrence in multivariate Cox regression analysis (HR, 4.451; 95% CI, 1.356–14.631; *p =* 0.014). This result is noteworthy in that few studies have linked CLNR with recurrence or DFS in patients with N1b PTC.

Cervical LN metastases are common in patients with PTC. If occult metastasis is included, cervical LN metastasis is reported in up to 90% of patients with PTC [22]. LN metastasis in patients with PTC matters because it is associated with a higher rate of recurrence after surgical treatment [23]. The ATA management guidelines published in 2015 classified patients with clinical N1 or >5 pathological N1 with all involved LNs < 3 cm in the largest dimension as having an intermediate risk of recurrence [15]. However, it is difficult to stratify patients with N1b PTC by risk using the existing criteria because the number of metastatic LNs depends on the extent of LN dissection [24].

The AJCC/UICC staging system is also insufficient to assess the risk of patients with N1b PTC; in the seventh edition, for patients > 45 years of age, N1a was classified as stage III and N1b as stage IV. In the revised version, all node-positive patients are classified as stage II, regardless of LN location in patients aged ≥55 years [12]. Although it had the effect of lowering the survival rate of high-stage patients [14], it is difficult to predict the prognosis in detail by observing patients with N1b PTC after treatment. To better predict prognosis in patients with N1b PTC, we introduced the concept of LNR, which is known to be a prognostic factor in various types of solid tumors [25,26,27].

LNR has attracted attention recently as a predictive marker for PTC recurrence that can complement the existing staging system or risk-stratification system. Yip et al. suggested that TNM nodal classification combined with LNR is a better predictor of recurrence than nodal classification alone in a retrospective cohort of 253 patients with PTC with LN metastasis [28]. Lee et al. reported that the eighth edition of the AJCC/UICC staging system or the 2015 ATA risk stratification had higher predictive power for recurrence in patients with PTC when combined with LNR [29]. Parvathareddy et al. retrospectively reviewed a cohort of 1407 patients with PTC and concluded that LNR predicted tumor recurrence better than the AJCC/UICC N stage (odds ratio, 1.96 vs. 1.30; *p*-value, 0.0184 vs. 0.3831) [30]. Kim et al. found, in a study with 745 patients with N1b PTC, that a lateral LNR > 0.3 was predictive of cancer-specific mortality [21]. If more evidence is accumulated, the LNR can be included in the staging system in the future.

Several studies have investigated cutoff values of LNR to verify its prognostic ability in patients with N1b PTC. Yuksel et al. reported that a cutoff of 0.21 for LNR was a predictor of DFS in patients with N1b PTC [31]. Lee et al. revealed that a cutoff of 0.218 for LNR was a predictor of recurrence in patients with N1b PTC [32]. Park et al. reported that an LNR > 0.22 significantly reduced loco-regional recurrence-free survival in patients with N1b PTC [33]. In this study, we found that the incidence of recurrence tended to be higher in patients with LNR ≥ 0.23 than in those with LNR < 0.23, but the difference was not significant (6.2% vs. 1.7%, *p =* 0.058).

Only a few studies have reported that CLNR in patients with N1b PTC had clinical significance. In a study of 324 patients with N1b PTC, a CLNR > 0.42 was an independent prognostic factor predicting loco-regional recurrence [34]. Ryu et al. described, in another study, that CLNR > 0.44 was associated with worse prognosis in patients with N1b PTC [35]. In our analysis, CLNR of 0.7 as a cutoff was statistically significant for predicting DFS. The differences in cutoff values of CLNR between studies may result from differences in the proportion of recurrent patients. Eleven (3.6%) patients developed recurrence in our study, whereas 14.5% and 21.5% of patients, respectively, developed recurrence in the other two studies mentioned above. Another reason why the cutoff values varied from study to study was the difference in the extent of LN dissection. The two studies cited above showed a smaller number of harvested LNs in the central compartment than in the present study. The number of harvested LNs in the central compartment was 11.0 ± 6.0 in one of those two previous studies [34] and 7.4 ± 6.0 in the other [35]. In contrast, our data demonstrated that 13.6 ± 7.6 LNs were harvested and examined in the central compartment. The mean values of LNs harvested in the central compartment were 13.2 in the low-LNR group and 14.3 in the high-LNR group. In addition, the number of positive central LNs was 9.2 ± 6.3 in the high-LNR group. These results show that more central LNs were harvested from the patients enrolled in this study compared with other studies.

In addition to CLNR, our multivariate Cox regression analysis results showed that perineural invasion was an independent prognostic factor for DFS. Perineural invasion refers to tumor cells circumferentially surrounding a nerve and is associated with an increased rate of recurrence and decreased survival, especially in head and neck cancers [36]. Rowe et al. reported that the perineural invasion of PTC was associated with extrathyroidal invasion [37]. To the best of our knowledge, no clinical study has revealed the relationship between perineural invasion and recurrence in thyroid cancer. A multicenter study in a large cohort should be performed to confirm the role of perineural invasion as a prognostic factor.

There were some limitations in this retrospective study that should be considered. This was a single-center study, and selection bias could have occurred, so our results may not be applicable to the broader population. Our results are difficult to generalize to patients with PTC because we only included pathologic N1b PTC. Other LN-related factors, such as extra-nodal extension or the maximal diameter of metastatic LNs, were not considered in our analysis, although they were reported to have prognostic value in previous studies [38,39]. We have a plan to perform a prospective study dealing with more LN-related factors in the future to overcome the limitations.

Our study has several strengths. All patients were diagnosed, treated, and followed up with according to a single standardized protocol. The relatively long period of postoperative follow-up also differentiated this study from others [34,35]. In addition, the metastatic and harvested LNs were evaluated and counted by a single pathology team, which increased the reliability of the data.

## 5. Conclusions

Our analysis indicated that CLNR was associated with recurrence in patients undergoing surgery after diagnosis with N1b PTC. CLNR ≥ 0.7 and perineural invasion were independent predictors of worse DFS. Notably, neither LNR in the whole neck nor LLNR was significant. This indicates that CLNR might be important in predicting recurrence with both N1a and N1b PTC. We expect this analysis to shed light on future investigations and to contribute to a better staging system or guidelines. In addition, this study provides evidence for which patients are more likely to experience recurrence after surgery. We recommend that attention should be given to patients with CLNR ≥ 0.7 after surgery for N1b PTC.

## Figures and Tables

**Figure 1 cancers-14-03677-f001:**
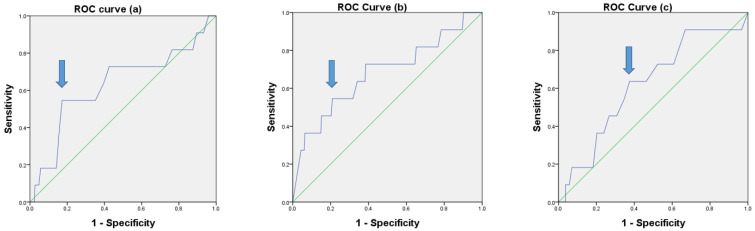
ROC curves for LNR (**a**), CLNR (**b**), and LLNR (**c**) (*p =* 0.141, *p =* 0.043, and *p =* 0.177, respectively).

**Figure 2 cancers-14-03677-f002:**
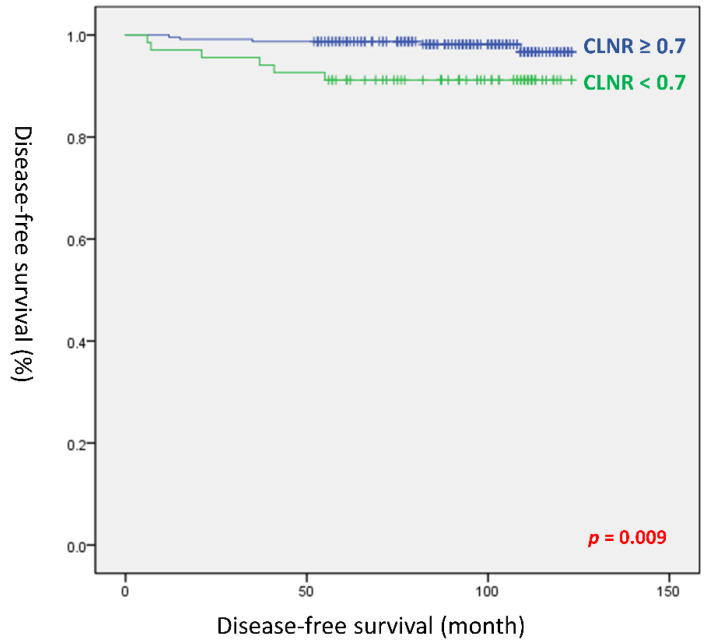
Disease-free survival curves according to the central lymph node ratio (log-rank *p =* 0.009).

**Table 1 cancers-14-03677-t001:** Comparison of baseline clinicopathological characteristics according to LNR (whole).

Cutoff 0.23	Low LNR (*n* = 178)	High LNR (*n* = 129)	*p*-Value
Age (years)	44.8 ± 12.6 (range, 15–73)	40.6 ± 14.1 (range, 15–78)	0.006
Female	123 (69.1%)	80 (62.0%)	0.222
Tumor size (cm)	1.5 ± 1.0 (range, 0.3–6.7)	1.8 ± 1.1 (range, 0.3–5.0)	0.004
Multifocality	97 (54.5%)	81 (62.8%)	0.161
Bilaterality	59 (33.1%)	48 (37.2%)	0.469
Gross ETE	32 (18.0%)	29 (22.5%)	0.385
Lymphatic invasion	124 (69.7%)	110 (85.3%)	0.002
Vascular invasion	8 (4.5%)	16 (12.4%)	0.016
Perineural invasion	12 (6.7%)	7 (5.4%)	0.811
BRAF positive	121/134 (90.3%)	75/94 (79.8%)	0.033
Harvested LNs	59.3 ± 22.6	53.7 ± 22.9	0.033
Central	13.2 ± 7.5	14.3 ± 7.6	0.234
Lateral	46.1 ± 19.3	39.5 ± 18.6	0.003
Positive LNs	8.0 ± 4.7	17.9 ± 9.3	<0.001
Central	3.6 ± 3.1	9.2 ± 6.3	<0.001
Lateral	4.4 ± 3.0	8.7 ± 5.6	<0.001
T stage			0.153
T1	124 (69.7%)	78 (60.5%)	
T2	18 (10.2%)	20 (15.5%)	
T3a	4 (2.2%)	2 (1.6%)	
T3b	26 (14.6%)	23 (17.8%)	
T4a	6 (3.4%)	6 (4.7%)	
TNM stage			0.081
Stage I	130 (73.0%)	106 (82.2%)	
Stage II	45 (25.3%)	21 (16.3%)	
Stage III	3 (1.7%)	2 (1.6%)	
Recurrence	3 (1.7%)	8 (6.2%)	0.058

Data are expressed as the patient’s number (%) or the mean ± standard deviation. A statistically significant difference was defined as *p <* 0.05. Abbreviations: LNR, lymph node ratio; ETE, extrathyroidal extension; BRAF, B-Raf proto-oncogene; LN, lymph node; T, tumor; TNM, tumor-node-metastasis.

**Table 2 cancers-14-03677-t002:** Comparison of baseline clinicopathological characteristics according to CLNR.

Cutoff 0.7	Low CLNR (*n* = 239)	High CLNR (*n* = 68)	*p*-Value
Age (years)	43.8 ± 13.6 (range, 15–78)	40.4 ± 12.3 (range, 22–77)	0.063
Female	170 (71.1%)	33 (48.5%)	0.001
Tumor size (cm)	1.5 ± 1.0 (range, 0.3–6.7)	2.0 ± 1.2 (range, 0.5–5.0)	0.001
Multifocality	140 (58.6%)	38 (55.9%)	0.781
Bilaterality	84 (35.1%)	23 (33.8%)	0.886
Gross ETE	48 (20.1%)	13 (19.1%)	1.000
Lymphatic invasion	178 (74.5%)	56 (82.4%)	0.200
Vascular invasion	15 (6.3%)	9 (13.2%)	0.073
Perineural invasion	15 (6.3%)	4 (5.9%)	1.000
BRAF positive	151/175 (86.3%)	45/53 (84.9%)	0.823
Harvested LNs	56.5 ± 22.8	58.8 ± 24.6	0.508
Central	13.9 ± 7.4	12.8 ± 8.0	0.303
Lateral	42.7 ± 19.4	45.7 ± 18.6	0.244
Positive LNs	10.3 ± 6.7	18.6 ± 10.7	<0.001
Central	4.6 ± 4.1	10.6 ± 6.9	<0.001
Lateral	5.7 ± 4.3	8.1 ± 5.8	<0.001
T stage			0.062
T1	164 (68.6%)	38 (55.9%)	
T2	24 (10.0%)	14 (20.6%)	
T3a	3 (1.3%)	3 (4.4%)	
T3b	38 (15.9%)	11 (16.2%)	
T4a	10 (4.2%)	2 (2.9%)	
TNM stage			0.210
Stage I	179 (74.9%)	57 (83.8%)	
Stage II	55 (23.0%)	11 (16.2%)	
Stage III	5 (2.1%)	0 (0.0%)	
Recurrence	5 (2.1%)	6 (8.8%)	0.017

Data are expressed as the patient’s number (%) or mean ± standard deviation. A statistically significant difference was defined as *p <* 0.05. Abbreviations: CLNR, central lymph node ratio; ETE, extrathyroidal extension; BRAF, B-Raf proto-oncogene; LN, lymph node; T, tumor; TNM, tumor-node-metastasis.

**Table 3 cancers-14-03677-t003:** Comparison of the baseline clinicopathological characteristics according to LLNR.

Cutoff 0.16	Low LLNR (*n* = 195)	High LLNR (*n* = 112)	*p*-Value
Age (years)	43.2 ± 12.9 (range, 15–77)	42.9 ± 14.2 (range, 15–78)	0.852
Female	127 (65.1%)	76 (67.9%)	0.707
Tumor size (cm)	1.5 ± 1.0 (range, 0.3–6.7)	1.8 ± 1.1 (range, 0.4–5.4)	0.074
Multifocality	102 (52.3%)	76 (67.9%)	0.008
Bilaterality	56 (28.7%)	51 (45.5%)	0.004
Gross ETE	35 (17.9%)	26 (23.2%)	0.299
Lymphatic invasion	138 (70.8%)	96 (85.7%)	0.003
Vascular invasion	10 (5.1%)	14 (12.5%)	0.027
Perineural invasion	10 (5.1%)	9 (8.0%)	0.332
BRAF positive	128/144 (88.9%)	41/53 (77.4%)	0.115
Harvested LNs	59.1 ± 22.5	53.1 ± 23.1	0.027
Central	13.5 ± 7.0	13.8 ± 8.5	0.736
Lateral	45.6 ± 18.9	39.3 ± 19.2	0.005
Positive LNs	9.3 ± 5.9	17.1 ± 10.1	<0.001
Central	5.3 ± 4.6	7.1 ± 6.6	0.010
Lateral	4.0 ± 2.5	10.0 ± 5.3	<0.001
T stage			0.642
T1	131 (67.2%)	71 (63.4%)	
T2	24 (12.3%)	14 (12.5%)	
T3a	5 (2.6%)	1 (0.9%)	
T3b	29 (14.9%)	20 (17.9%)	
T4a	6 (3.1%)	6 (5.4%)	
TNM stage			0.840
Stage I	152 (77.9%)	84 (75.0%)	
Stage II	40 (20.5%)	26 (23.2%)	
Stage III	3 (1.5%)	2 (1.8%)	
Recurrence	4 (2.1%)	7 (6.3%)	0.106

Data are expressed as the patient’s number (%), or mean ± standard deviation. A statistically significant difference was defined as *p <* 0.05. Abbreviations: LLNR, lateral lymph node ratio; ETE, extrathyroidal extension; BRAF, B-Raf proto-oncogene; LN, lymph node; T, tumor; TNM, tumor-node-metastasis.

**Table 4 cancers-14-03677-t004:** Univariate and multivariate analyses of the risk factors for recurrence.

	Univariate		Multivariate	
	HR (95% CI)	*p*-Value	HR (95% CI)	*p*-Value
Tumor size	1.707 (1.176–2.476)	0.005		
Vascular invasion	4.320 (1.145–16.302)	0.031		
Perineural invasion	5.588 (1.482–21.068)	0.011	6.045 (1.593–22.937)	0.008
Positive LNs (whole)	1.048 (1.009–1.090)	0.017		
Positive LNs (lateral)	1.095 (1.006–1.191)	0.037		
CLNR	11.026 (1.242–97.862)	0.031		
CLNR < 0.7	Ref.		Ref.	
≥0.7	4.238 (1.292–13.896)	0.017	4.451 (1.356–14.613)	0.014
TNM stage				
I	Ref.			
II	0.382 (0.048–3.017)	0.362		
III	10.094 (1.262–80.753)	0.029		

Data are expressed as the hazard ratio and 95% confidence interval. A statistically significant difference was defined as *p <* 0.05. Abbreviations: HR, hazard ratio; CI, confidence interval; LN, lymph node; CLNR, central lymph node ratio; TNM, tumor-node-metastasis.

**Table 5 cancers-14-03677-t005:** Recurrence patterns for the study population.

	No. of Patients	Age	Sex	Tumor Size (cm)	Recurrence Site	DFS (Months)
Low CLNR	1	73	Male	2.5	Contralateral Level-3, -4, -5 LNs	109
	2	30	Female	1.3	Ipsilateral Level-3 LNs	15
	3	68	Female	4.5	Contralateral Strap muscle	82
	4	24	Female	1.0	Ipsilateral Level-6 LNs	35
	5	27	Female	1.7	Ipsilateral Level-4 LNs	12
High CLNR	1	49	Male	5.0	Level-5 LNs, left	55
	2	37	Male	3.7	Contralateral Level-2, -3 LNs	21
	3	32	Male	4.0	Contralateral Level-3, -4 LNs	41
	4	35	Female	1.1	Lung, left lower	37
	5	38	Male	2.5	Level-6 LNs, left	7
	6	29	Female	2.7	Contralateral Level-2, -4, -5 LNs	8

Abbreviations: DFS, disease-free survival; CLNR, central lymph node ratio; LN, lymph node.

## Data Availability

The data that support the findings of this study are available on request from the corresponding author. The data are not publicly available due to privacy or ethical restrictions.

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
