# Peer review of "Central Lymph Node Ratio Predicts Recurrence in Patients with N1b Papillary Thyroid Carcinoma"

_cancers, 2022, doi:10.3390/cancers14153677_

Round 1

Reviewer 1 Report

Dear authors,

your study aim to investigate the relationship between LNR and disease-free survival (DFS) in PTC patients with lateral LN metastases (N1b PTC). This is an interesting paper, but some minor revisions are need before its publication in Cancer journal

Introduction: the introduction is too short, I suggest to deepen the topic by adding more recent evidence of literature.

Discussion: what are the limitations of this research? how these can be overcome?

Conclusions: the conclusions must be rewritten trying to better finalize the data of the research carried out.

.

Reviewer 2 Report

This is a retrospective study evaluating the contribution of central and lateral neck compartment lymph node ratios for predicting outcomes for PTC. The study evaluated over 300 subjects and as such is smaller than some of the previous reports on similar issues. The lymph node ratios (LNR) reported vary minimally from previous reports but there is new information regarding the independent contributions of each nodal compartment.

The study is well designed and the manuscript is well written without any major or minor corrections needed.

Reviewer 3 Report

The manuscript provides some additional data on prognostication in papillary thyroid carcinoma, particularly in the context of TNM staging system limitations.

The paper is well written. The data are well presented; the results are thoroughly discussed and justify the conclusions.

There are, however, a few issues to resolve before the manuscript's publication.

The LNR, CLNR and LLNR cut-off values applied by the authors should be justified. The authors have stated that the ROC analysis has been performed, and the appropriate parameters have been calculated - however, the results are not presented in the paper. Those results should be definitely included. 

The minor issues:

- the version of  the AJCC TNM staging system should be added (lines 42-43)

- line 86: histological should be changed to cytological (as the FNAB was performed to confirm neck recurrence)

- line 215: predicative should be changed to predictive.
